# Image-Based Artificial Intelligence Methods for Product Control of Tablet Coating Quality

**DOI:** 10.3390/pharmaceutics12090877

**Published:** 2020-09-15

**Authors:** Cosima Hirschberg, Magnus Edinger, Else Holmfred, Jukka Rantanen, Johan Boetker

**Affiliations:** 1BASF A/S, Malmparken 5, 2750 Ballerup, Denmark; cosima.hirschberg@basf.com; 2Faculty of Health and Medical Sciences, University of Copenhagen, 2100 Copenhagen, Denmark; prf298@alumni.ku.dk (M.E.); jukka.rantanen@sund.ku.dk (J.R.); 3Research Group for Nano-Bio Science, National Food Institute, Technical University of Denmark, Kemitorvet, 2800 Kgs. Lyngby, Denmark; elshol@food.dtu.dk

**Keywords:** in silico modelling, neural networks, image analysis, artificial intelligence, multivariate analysis

## Abstract

Mimicking the human decision-making process is challenging. Especially, many process control situations during the manufacturing of pharmaceuticals are based on visual observations and related experience-based actions. The aim of the present work was to investigate the use of image analysis to classify the quality of coated tablets. Tablets with an increasing amount of coating solution were imaged by fast scanning using a conventional office scanner. A segmentation routine was implemented to the images, allowing the extraction of numeric image-based information from individual tablets. The image preprocessing was performed prior to utilization of four different classification techniques for the individual tablet images. The support vector machine (SVM) technique performed superior compared to a convolutional neural network (CNN) in relation to computational time, and this approach was also slightly better at classifying the tablets correctly. The fastest multivariate method was partial least squares (PLS) regression, but this method was hampered by the inferior classification accuracy of the tablets. Finally, it was possible to create a numerical threshold classification model with an accuracy comparable to the SVM approach, so it is evident that there exist multiple valid options for classifying coated tablets.

## 1. Introduction

Visual observation is still a commonly used method for the characterization of pharmaceutical systems. One example of this is the expression ‘cake appearance’ that is commonly used to describe freeze-dried products [1]. Subjective observations of these products are used to categorize them according to different levels of, e.g., collapse, cracking, fracturing, shrinkage. Similarly, the quality of a coated tablet can be evaluated based on visual observation, and terms like orange peeling, cracking, and erosion, are commonly used to describe defects in coated tablets. In many cases, an experienced pharmaceutical scientist and/or process operator can identify these defects with a plain eye and important decision-making for process control purposes can be performed based on these visual observations. This is challenging to document and missing an experienced person with key product/process understanding can be detrimental. Robust algorithms and computational methods [2,3] capable of mimicking human decision-making processes will be especially important considering the fast development of imaging tools.

The quality control of a coating layer is especially important for the functional coatings. Different tools and techniques have been developed to track the coating process [4]. The most common form of coating used in pharmaceutical manufacturing is film coating, where a solution is slowly sprayed on a bed of moving tablets [5]. By tracking the volume of coating solution and calculating the total surface area of the tablets, the average amount of coating on each tablet can be estimated. However, direct evaluation of the distribution of the coating liquid on tablets is challenging. Process analytical technology (PAT) has been successfully implemented to track the coating process, both in-line and on-line modes, with high precision. Near-infrared [6,7,8], conventional computer scanners [9], terahertz pulsed imaging [10], and Raman spectroscopy [11,12,13,14,15,16,17] have been successfully used to quantify the coating thickness. However, these techniques are typically capable of measuring an average value of the coated tablet and often do not show the coating distribution on the individual tablet. Recent technical development of process analytical tools allows for hyperspectral imaging of coated tablets and these techniques can capture nearly 100% of the tablets, but at the same time they create very large datasets [18,19,20]. In silico simulations of coating processes have also been in the focus of research in recent years [21,22,23].

Different data processing methods have been used to analyze data from the PAT interfaces, especially principal component analysis (PCA) and partial least square (PLS) regression [16,24]. The use of PCA and PLS for multivariate image analysis of the coating uniformity of tablets has also been successfully performed in the literature [25]. With increasing data complexity, these approaches might not be powerful enough to find the underlying patterns. Machine learning based on the use of support vector machines (SVM) and convolutional neural networks (CNN) provides possibilities for processing complex datasets with different inputs. The SVM [26,27] and CNN [28] methods have previously been shown to display great potential in characterizing information from images. Machine learning and the use of neural networks provide possibilities for the processing of large datasets with different inputs, as well as for identifying non-linear interactions in the data set. These artificial networks can be used for predictive modelling, adaptive control, and applications, where they can be trained using dedicated training datasets.

Behind the neural network exists a computational model for information processing, usually using an adaptive system, which changes its structure based on the information within the network. Neural networks are usually using non-linear data modelling or decision-making tools. The CNN method relies on a computational model for information processing and this computational model has an adaptive nature, which changes its structure based on the information given to the model. The CNN method can be used to find and model complex underlying relationships between the inputs and outputs that aim to find hidden patterns. For a deeper description of the CNN approach, the reader is referred to the literature [29]. Neural networks can potentially be used for a wide array of pharmaceutical applications ranging from early drug discovery to quality control of the final product. SVM methods that rely on an optimal hyperplane, which is defined as the hyperplane with the maximal margin of separation between two given classes, can be constructed [30]. For a description of how the SVM method transforms the data, the reader is referred to such work as Hearst et al. [30] and Noble [31].

The aim of the present work was to investigate the use of image analysis to classify the quality of coated tablets.

## 2. Material and Methods

### 2.1. Materials

For this work, four batches of tablet cores (Table 1) were coated with a tartrazine colored coating liquid (Table 2) for various periods of time.

Tablets were analyzed either as uncoated or coated with a coating speed of 19 g/min for approximately 3 min (slightly coated), 7 min (moderately coated), or 20 min (fully coated) on a fluid bed coater (Combi Coata, Model CC1/LAB, Niro Atomizer, Denmark). The coating was performed using an inlet temperature of 58 °C, nozzle pressure of 0.7 bar, and preheating of the tablets to 45 °C (approx. 5 min).

### 2.2. Methods

#### Scanning

The four different tablet batches, containing a total of 2318 tablets, were imaged on a Bizhub C360 scanner (Konica Minolta, Tokyo, Japan), with a resolution of 600 dpi and width × height of 9921 × 7015 pixels. The uncoated batch consisted of 623 tablets, the slightly coated batch contained 596 tablets, the moderately coated batch contained 595 tablets and the fully coated batch had 504 tablets.

## 3. Segmentation and Computational Analysis of Tablets

Image segmentation was performed in MATLAB^®^ (MathWorks, Natick, MA, USA) using the function bw boundaries [32]. PLS and SVM discriminative analysis models were calculated without prepressing using the PLS toolbox (Eigenvector Research Inc., Manson, WA, USA) with venetian blinds cross-validation and downsampled with a factor of 50 in order to increase the computational speed. The PLS and SVM models utilized the default parameter values. The numerical threshold classification (NTC) method was performed in MATLAB^®^ (Natick, MA, USA) on the 50 times downsampled images by performing an intensity summation over the spatial pixel positions for the single tablet images. The summed intensities were subsequently subdued to a threshold function that would act as a classifier. The CNN algorithm used in this paper was based on the MathWorks deep learning network for classification [33]. For the CNN algorithm, 372 single tablet training images were randomly selected for each of the four batches and the number of training epochs was 128 with 11 iterations per epoch and a maximum number of iterations of 1408. The sizes of all max-pooling filters and convolutional filters were 2 × 2 and 3 × 3, respectively, and the data were shuffled for each epoch. An epoch is defined as a full training cycle on the entire data training set. The max-pooling filters are used to further downsample the data, thus making it possible to increase the number of filters without increasing the number of computations needed. The max-pooling filter returns the maximum value in each rectangular region of inputs. The datasets and code snippet can be retrieved from https://sid.erda.dk/public/archives/fa29ef515bd7b0db948c58f9d74cfa93/published-archive.html.

## 4. Results and Discussion

The scanned images of both uncoated tablets and tablets with an increasing amount of coating solution are presented in Figure 1. It is directly apparent that with increasing coating deposition on the tablets a more uniform yellow color can be observed.

The proposed data analysis approach consists of two major steps. The first step is a tablet segmentation method that identifies every single tablet in the scanned image. This segmentation procedure is performed to facilitate the possibility of analysing the tablets on an individual level. The segmented tablets in the original image are subsequently stored as individual image files (Figure 2C).

The tablet segmentation algorithm allows the individual images to be parsed to the analytical models. The individual tablets in the four tablet batches can subsequently be characterized by their membership to their corresponding class where the uncoated batch is assigned to class 1, the slightly coated batch to class 2, the moderately coated batch to class 3, the fully coated batch to class 4, and the tablets not strictly predicted into any class are assigned to class 0. The PLS model regression can obtain a model that correctly classifies 2237 of the 2297 tablets and hence was unable to classify 60 tablets (Figure 3). This leads to an accuracy of correctly classified tablets of 97.4% for the PLS model. The SVM method is also capable of providing a method for calculating the predicted class and, using this method, it is observed that three tablets are misclassified, leading to a 99.9% accuracy of the model. The CNN approach was capable of calculating the predicted class with an accuracy of 99.6%, thus having misclassified a total of nine tablets out of the 2297 tablets (Figure 3).

It is hence evident that all the three methods can correctly classify the predominant portion of the tablets where the SVM machine learning approach achieves the highest amount of correctly classified tablets. A reason why the PLS approach performs rather well is that this is a highly linear problem, where the tablets are either more or less yellow and this linearity of the data responses is captured by the linear nature of the PLS regression [34,35]. This changing intensity in the yellow hue has a direct impact on the intensity values in the blue color channel where white tablets have high values, and increasingly yellow tablets have declining values in the blue color channel. This univariate variation in only the blue color channel is of course a property that is related to the yellow color of the coating liquid. In case a differently colored coating liquid was utilized, a different color channel or a combination of color channels may have to be used instead. These declining values in the blue color channel can be directly visualized by performing the previously described intensity summation over the spatial pixel positions for the single tablet images and subsequently performing an NTC of the summed intensity values (Figure 4). The summed intensities also indicate that there is a color intensity value distribution within each class and such distributions will instigate that misclassifications can occur.

The NTC method misclassifies three tablets and it hence has a model accuracy of 99.9%, which is identical to the SVM method. It is evident that a linear threshold method would provide an adequate approach for correctly classifying the differently coated tablet entities for this dataset. It should be highlighted that the datasets for the PLS, SVM, CNN, and the NTC models are not identical as the PLS, SVM, and CNN models contained both the red, green, and blue channels, whereas the NTC model only contained the summation of the highly relevant blue channel. The inclusion of only the relevant variable in the NTC method will of course provide a positive impact on the model robustness and model outcome [36,37]. This can be evidenced by performing an SVM analysis on only the blue channel that yields a model with an accuracy of 100% (data not shown).

The ordeal in this tablet coating color intensity quantification task is therefore to correctly identify the appropriate data analytical approach that provides enough specificity while not being too computationally costly and time-consuming. Here, the NTC method on selected variables is by far the most computationally efficient approach, providing the same accuracy of classification as the SVM method on the full dataset. Therefore, knowing a priori which part of the data contains the relevant information (in this case, the blue color channel) can also assist in improving model accuracy. This a priori knowledge is, however, not always available when a new dataset is investigated. Trial and error analysis of which algorithm best fits the purpose and the given dataset is thus always needed and general applicability cannot be asserted for a given method. As an example, the NTC method would be an ill conditioned option for a multivariate problem.

## 5. Conclusions

This study demonstrated the possibility of using a conventional office scanner for quality control of coated tablets. A critical part of this analysis was the segmenting of the tablets in the scanned images into individual files containing the single tablets. The results from the image analysis segmentation routines were used to create both linear regression models alongside machine learning and deep learning models for classifying the coating quality of coated tablets. The SVM technique performed superior compared to CNN in relation to computational time and was also slightly better at classifying the tablets correctly. The fastest multivariate method was PLS, but this method was also hampered by inferior classification accuracy of the tablets. Finally, it was possible to create an NTC model with similar accuracy to the SVM approach, so it is evident that there exist multiple valid options for classifying the coated tablets. The selection of an algorithm (NTC, PLS, SVM, CNN, etc.) may thus be based on a trial and error analysis of which algorithm best fits the purpose and the given dataset.

## Figures and Tables

**Figure 1 pharmaceutics-12-00877-f001:**
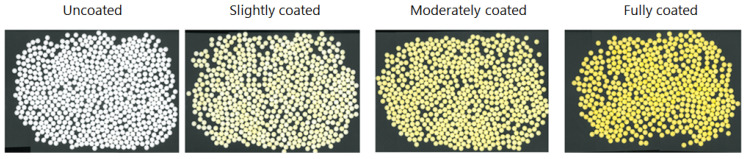
The four different batches with an increasing amount of coating solution.

**Figure 2 pharmaceutics-12-00877-f002:**
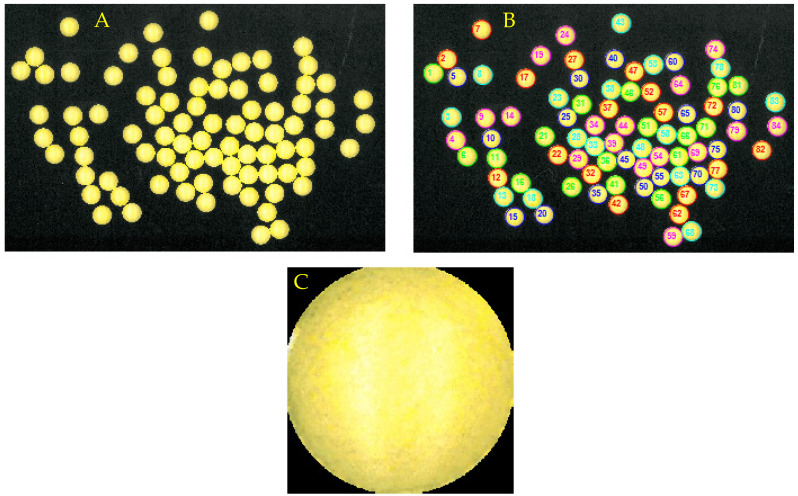
Scanned image of the coated tablets (**A**), segmentation of the individual coated tablets in the scanned image (**B**), and an example of a segmented image of one of the tablets (**C**).

**Figure 3 pharmaceutics-12-00877-f003:**
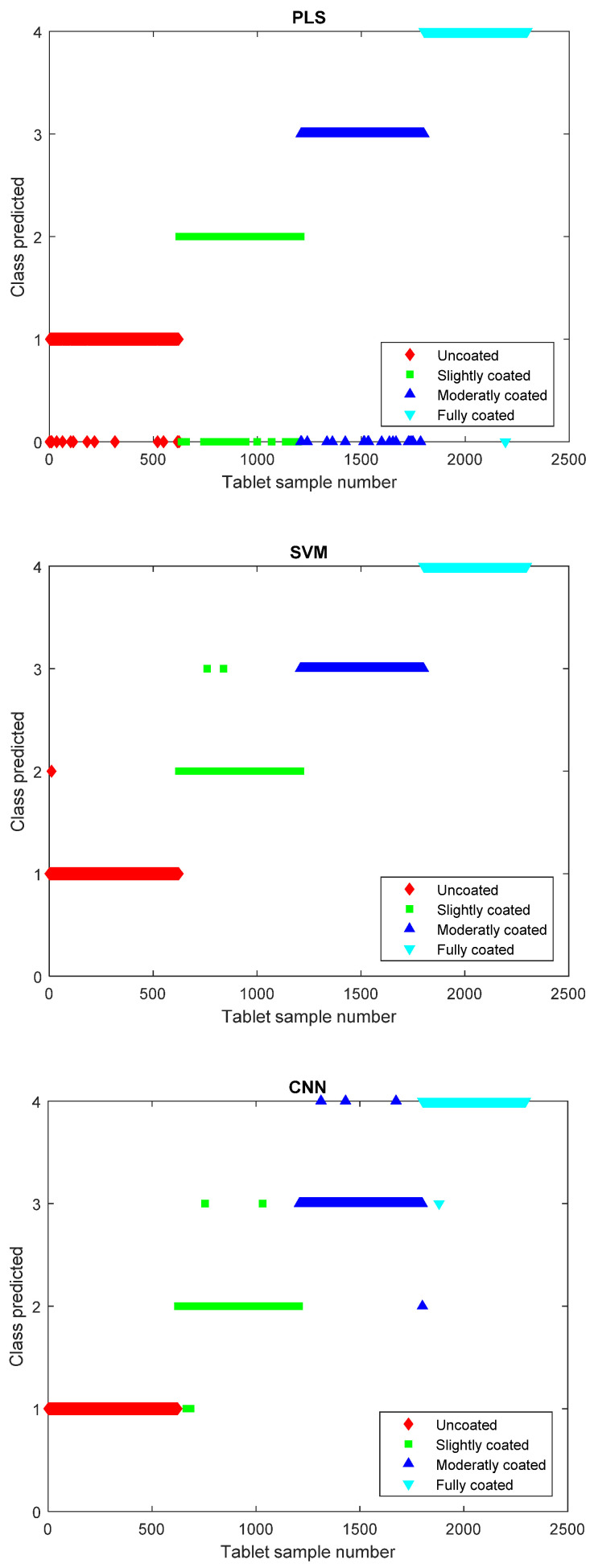
Partial least squares (PLS) model predicted class versus tablet sample number (**top**). Support vector machine (SVM) model predicted class versus tablet sample number (**middle**). Convolutional neural network (CNN) model predicted class versus tablet sample number (**bottom**).

**Figure 4 pharmaceutics-12-00877-f004:**
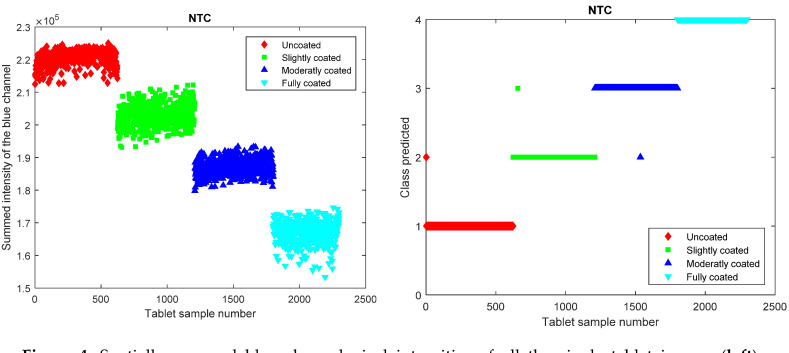
Spatially summed blue channel pixel intensities of all the single tablet images (**left**). Numerical threshold classification method showing predicted class versus tablet sample number (**right**).

**Table 1 pharmaceutics-12-00877-t001:** Tablet composition.

Constituent	%(*w*/*w*)
Emcompress	57
Avicel PH102	38
Talcum and magnesium stearate (9 + 1)	5

**Table 2 pharmaceutics-12-00877-t002:** Coating liquid composition.

Constituent	%(*w*/*w*)
Tartrazine	0.05
Ponceau-4R	0.05
Glycerol 85%	0.53
Water	99.37

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
