# Peer review of "Image-Based Artificial Intelligence Methods for Product Control of Tablet Coating Quality"

_pharmaceutics, 2020, doi:10.3390/pharmaceutics12090877_

Round 1

Reviewer 1 Report

Thank you for responding to my comments and making the minor corrections.
I am disappointed that the authors have not considered the fact that the red (R) and the green (G) colours must be included in the analysis. The explanation in the text (lines 154-164) neglects the fact that a low value of B does not mean the colour will be yellow. As I said, RGB values for black, red, lime/green, maroon and olive, all have B=0. It is a 3-dimension problem.
The robustness of the method needs to be addressed.
Line 158 – channel not cannel.

Reviewer 2 Report

The authors addressed most of my comments satisfactorily.

Author Response

A spell check has been performed in the revised manuscript. 

Round 2

Reviewer 1 Report

Authors:
In terms of general colour classification, the red and green are not irrelevant, they might not be critical for this particular case. An explanation of the selection criteria with appropriate reference(s) is required.
The argument about robustness is weak. The sources for variability are not only the amount of coating liquid and it was not shown that “the biggest source for variation would perhaps be the added amount of coating liquid”. The authors would have to identify all the critical parameters of the coating process and evaluate / rank them.
The critical factors related to the scanner, or types of scanners, their light source, etc. were not mentioned in the manuscript. Robustness of the process and method of analysis are more complex than simply considering the amount of coating liquid.

Author Response

In terms of general colour classification, the red and green are not irrelevant, they might not be critical for this particular case. An explanation of the selection criteria with appropriate reference(s) is required.

Variable selection and the reason why to utilize variable selection has already been described in line 176. An extra reference has been inserted here.

The argument about robustness is weak. The sources for variability are not only the amount of coating liquid and it was not shown that “the biggest source for variation would perhaps be the added amount of coating liquid”. The authors would have to identify all the critical parameters of the coating process and evaluate / rank them.
The critical factors related to the scanner, or types of scanners, their light source, etc. were not mentioned in the manuscript. Robustness of the process and method of analysis are more complex than simply considering the amount of coating liquid.

It is here important to look at what this paper is about. Is it about the robustness of a particular method setup using an office scanner? Or is it about comparing image based-artificial intelligence methods? It is very much about comparing image based-artificial intelligence methods and not about the robustness of a given hardware setup. This is directly seen in figure 3 and 4 where we are comparing algorithms (PLS, SVM, CNN and NTC) on the same dataset.  

The suggestion of: “identify all the critical parameters of the coating process and evaluate / rank them” would be out of scope for this paper, as this paper is not at all about identifying critical parameters of a given laboratory scale coating equipment. The same argumentation goes for the hardware of the scanner as this paper is dealing with the selection of an appropriate algorithm that best fits the purpose and the given dataset and this is hence not limited to a given production technique or analytical setup. Finally, would ranked lists of critical parameters change any of the conclusions in this manuscript? The answer is no.                 

This manuscript is a resubmission of an earlier submission. The following is a list of the peer review reports and author responses from that submission.

Round 1

Reviewer 1 Report

Hirschberg et al. present an approach to automatically classfiy individual tablets according to the degree of being coated. To this end, they have tested different ML algorithms and compared their performance. While I commend the authors on the overall approach, I find the manuscript wanting in several aspects:

  • The presentation of the results leaves a lot to be desired. In general the degree of detail in the methods section is insufficient, e.g.: How were the hyper-parameters set for the SVM? Where there hold-out sets used or are the reported accuracies in respect to the whole dataset? Why was subsampling applied for the CNN training? Which pretrained network was used? ...
  • Even if no full-fledged software is produced, code snippets, datasets and models should be made available. Otherwise, it is borderline impossible to assess / apply the approach presented.
  • As the authors point out the classification problem at hand is very nicely behaved and linearly separable. To prove general applicability of the approach, at least one other (maybe more challenging task) should be considered.
  • The graphs should be better annotated (e.g. Figure 3).
  • Missing analysis of the wrongly classified observations.
  • The authors claim that "The aim of this work was to provide an image segmentation approach for comparing PLS, SVM, CNN, and numerical threshold methods abilities for classifying the quality of color-coated tablets from scanned images of tablet batches.", but it appears the segmentation algorithm is not part of the original work presented but simply a pre-requisite.

Reviewer 2 Report

 The article is acceptable in the current format as the proposed method can classify the tablets to a high degree off accuracy and with a low mis classification rate.

Reviewer 3 Report

 Image-based artificial intelligence methods for product control of tablet coating quality

The paper provides some contribution to the scientific community in the area of image analysis to classify the quality of coated tablets.

As stated on the paper, the aim of the work was to provide an image segmentation approach for comparing PLS, SVM, CNN, and numerical threshold methods abilities for classifying the quality of color-coated tablets from scanned images of tablet batches.

My main concerns are:

  • the authors used blue for the classification and in my view, they should have used red and green as well as blue. For a 1-byte code, the RGB values for yellow are (255,255,0), so, provided that R and G are maximum, then blue has information on how dark is the yellow, going from B=0 for pure yellow to B=255 for pure white. But it seems that the authors simply ignore R and G. The RGB values for Black, red, lime/green, maroon and olive, all have B=0. Please comment on the rationale for the selection of blue alone.
  • How robust is this system? If the same coating conditions are used by other researchers from a different lab, would they get the same results? If another scanner (with similar specifications) is used, would the results be the same? If the system has been trained on one scanner, the unseen data should come from another scanner, to guarantee that no bias / overtraining has happened.

Minor points:

  • Figure 3: all figures on the left side (misclassified tablets) are redundant. The same information is provided on the class predicted plots.
  • Line 138 The tablet segmentation algorithm allows the individual images can be to parsed to the analytical model.
  • Line 144 Please define accuracy